

# Diversity and evolution of the endosymbionts of *Bemisia tabaci* in China

Xiao-Tian Tang[1,2,*], Li Cai[1,*], Yuan Shen[3] and Yu-Zhou Du[1,4]

[1] School of Horticulture and Plant Protection & Institute of Applied Entomology, Yangzhou University, Yangzhou, Jiangsu, China
[2] Department of Entomology, Texas A&M University, College Station, TX, USA
[3] Agriculture and Forestry Bureau of Binhu District, Wuxi, China
[4] Joint International Research Laboratory of Agriculture and Agri-Product Safety, the Ministry of Education, Yangzhou University, Yangzhou, Jiangsu, China
* These authors contributed equally to this work.

## ABSTRACT

The whitefly *Bemisia tabaci* (Gennadius) (Hemiptera: Aleyrodidae) is a cryptic species complex, including members that are pests of global importance. This study presents a screening of *B. tabaci* species in China for infection by the primary endosymbiont, *Portiera aleyrodidarum*, and two secondary endosymbionts, *Arsenophonus* and *Cardinium*. The results showed that *P. aleyrodidarum* was detected in all *B. tabaci* individuals, while *Arsenophonus* was abundant in indigenous species of *B. tabaci* Asia II 1, Asia II 3, and China 1 but absent in the invasive species, Middle East-Asia Minor 1 (MEAM1); *Cardinium* presented in the Mediterranean (MED), Asia II 1 and Asia II 3 species but was rarely detected in the MEAM1 and China 1 species. Moreover, phylogenetic analyses revealed that the *P. aleyrodidarum* and mitochondrial cytochrome oxidase 1 (*mtCO1*) phylograms were similar and corresponding with the five distinct cryptic species clades to some extent, probably indicating an ancient infection followed by vertical transmission and subsequent co-evolutionary diversification. In contrast, the phylogenetic trees of *Arsenophonus* and *Cardinium* were incongruent with the *mtCO1* phylogram, potentially indicating horizontal transmission in *B. tabaci* cryptic species complex. Taken together, our study showed the distinct infection status of endosymbionts in invasive and indigenous whiteflies; we also most likely indicated the co-evolution of primary endosymbiont and its host as well as the potential horizontal transfer of secondary endosymbionts.

# INTRODUCTION

*Bemisia tabaci* is a cryptic species complex comprising a minimum of 40 morphologically similar species (*De Barro et al., 2011*; *Dinsdale et al., 2010*; *Hu et al., 2018*; *Wang, Li & Liu, 2017*). Among members of the complex, the Middle East-Asia Minor 1 (MEAM1) and Mediterranean (MED) groups (commonly known as the B and Q biotypes, respectively) have drawn much attention due to their global invasion and vectoring

Corresponding author
Yu-Zhou Du, yzdu@yzu.edu.cn

important plant pathogens (e.g., tomato yellow leaf curl virus) (*Brown, 1994*; *Cohen & Nitzany, 1966*). In China, *B. tabaci* was not considered as a serious pest until the arrival of the MEAM1 group in the mid-1990s (*Qiu et al., 2007*). In 2004, the MED group was detected, and rapidly became widely distributed in China, causing considerable damage to a wide range of vegetables, fibers, and ornamental crops (*Chu et al., 2005*). It is interesting that MED has been replacing the earlier invader MEAM1 as well as several indigenous species of whiteflies (e.g., Asia II and China 1) in many regions (*Liu et al., 2007*; *Sun et al., 2013*).

Associations between insects and endosymbionts are quite common in nature. It has been estimated that at least 15–20% of all insect species live in symbiotic relationships with bacteria (*Douglas, 1998*; *Gosalbes et al., 2010*). Endosymbionts associated with insects can be classified into primary endosymbionts (P-endosymbionts) and secondary endosymbionts (S-endosymbionts) (*Baumann, 2005*). The P-endosymbionts are obligate and usually have mutualist relationships with their hosts. Besides, P-endosymbionts are generally localized in bacteriocytes of bacteriome (*Caspi-Fluger et al., 2011*) and transmitted vertically from mother to progeny (*Werren & O'Neill, 1997*). In contrast, the S-endosymbionts are usually facultative symbionts, and they could reside in several host tissues such as gut, hemolymph, Malpighian tubules, salivary glands or ovarian cells (*Cicero, Fisher & Brown, 2016*; *Cooper, Sengoda & Munyaneza, 2014*; *Dobson et al., 1999*; *Zchori-Fein, Roush & Rosen, 1998*). Infection of secondary endosymbionts can be either maternally inherited or horizontally transmitted (*Moran & Baumann, 2000*). It has been discovered that the P-endosymbionts *Portiera aleyrodidarum* and S-endosymbionts such as *Arsenophonus*, *Cardinium*, *Fritschea*, *Hamiltonella*, *Rickettsia*, and *Wolbachia* were infected in whiteflies (*Bing et al., 2013*; *Chiel et al., 2007*; *Chu et al., 2011*; *Everett et al., 2005*; *Karut et al., 2017*; *Thao & Baumann, 2004*; *Zchori-Fein, Lahav & Freilich, 2014*). Previous studies have investigated the prevalence, diversity and evolution of endosymbionts in the *B. tabaci* species complex from different countries or regions (e.g., Turkey, China, Brazil, and Africa) (*Ahmed et al., 2010*; *Bing et al., 2013*, *2014*; *Ghosh, Bouvaine & Maruthi, 2015*; *Hashmi et al., 2018*; *Jahan et al., 2015*; *Karut et al., 2017*; *Marubayashi et al., 2014*; *Santos-Garcia et al., 2015*; *Sseruwagi et al., 2018*; *Thierry et al., 2011*, *2015*). However, most of these reports only focused on the two invasive cryptic species MEAM1 and MED; furthermore, for studies from China, the sample size and distribution range were limited. For example, regarding prevalence of S-endosymbionts in *B. tabaci*, only the laboratory samples from two provinces were investigated (*Bing et al., 2013*); only one S-endosymbiont (*Wolbachia*) was explored from a slightly larger geographical scale (*Bing et al., 2014*).

Therefore, although substantial datasets regarding endosymbionts prevalence of *B. tabaci* species complex are present, their current situation in whole China is still not very clear and further investigation is essential. In the present study, our first goal is to investigate the prevalence and diversity of the P-endosymbionts *P. aleyrodidarum* and two common S-endosymbionts *Arsenophonus* and *Cardinium* (as representatives of S-endosymbionts) within *B. tabaci* in a wider range of China. We also aimed to explore the evolutionary relationships between these three endosymbionts and their host based on
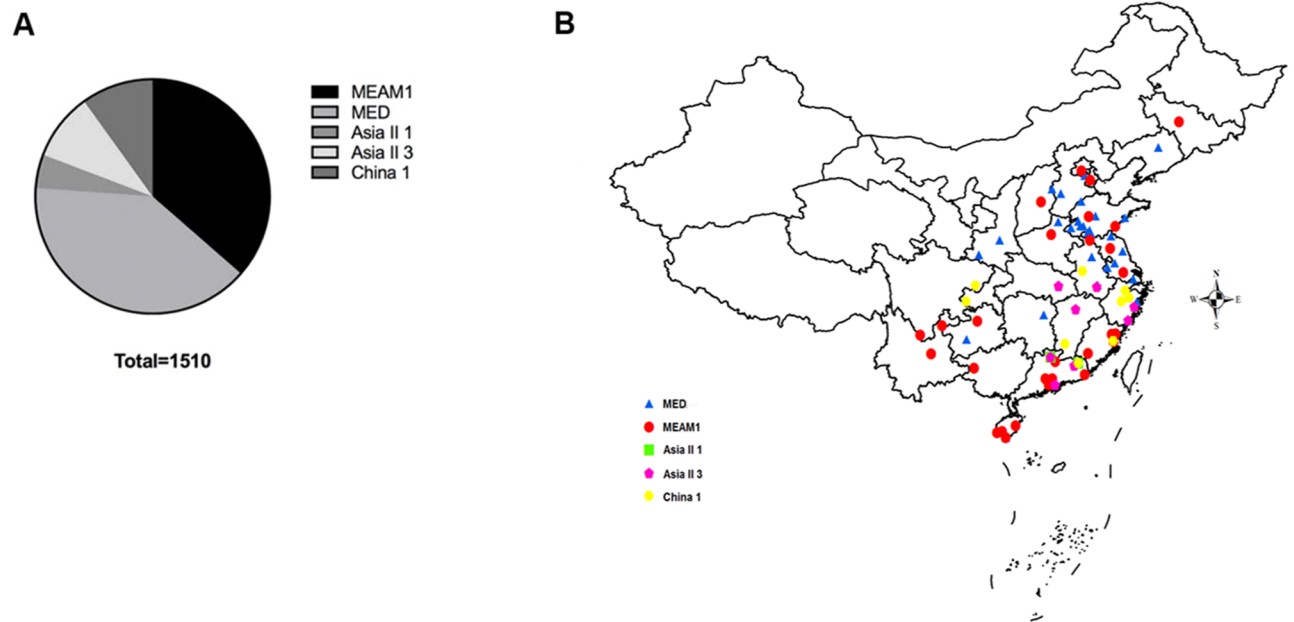

**Figure 1 The quantity and distribution of *B. tabaci* cryptic species in China.** (A) The quantity of each *B. tabaci* cryptic species based on molecular identification. (B) The locations of the *B. tabaci* cryptic species populations in China. Names of locations are given in Table S1. Maps were created using Esri's ArcGIS platform (http://www.esri.com/software/arcgis).

phylogenetic analyses of 16S and 23S ribosomal DNA (*rDNA*) (from endosymbionts) as well as mitochondrial cytochrome oxidase 1 (*mtCO1*) gene (from *B. tabaci*). Our study will not only uncover the current status of endosymbionts infection within *B. tabaci* in China, but also greatly provide a supplement for studies of *B. tabaci* endosymbionts worldwide.

# MATERIALS AND METHODS

## Sample collection

A total of 1,510 *B. tabaci* individuals were collected from 71 geographical locations, including 19 provinces and four municipalities in China (Fig. 1). At each location, *B. tabaci* samples were collected from different leaves of separate plants. The collection details, geographical sites and host plants were described in Table S1.

## DNA extraction and gene amplification

Total DNA was extracted from individual whitefly as described in *Luo et al. (2002)*. The primers of *mtCO1* gene was used for whitefly species identification. 16S *rDNA* primers were used to detect *P. aleyrodidarum* and *Cardinium*, and for *Arsenophonus*, 23S *rDNA* primers were utilized. The primers, annealing temperature, and predicted PCR products size were shown in Table 1. The Polymerase chain reaction (PCR) reaction mixture contained one U Taq DNA polymerase, five μl (10×) reaction buffer, three μl MgCl₂ (final concentration of 25 mmol/l), two μl dNTPs (10 mmol/l), two μl of forward and reverse primers (20 μmol/l each) and two μl of template DNA (*Simon et al., 1994*). PCR products were visualized by 1.5% agarose gels
**Table 1 The sequences, annealing temperature, product size and references of primers used in this study.**

| Gene | Primer sequence (5′→3′) | Annealing temperature | Product size (bp) | Reference |
|---|---|---|---|---|
| *B. tabaci mtCOI* | C1-J-2195: TTGATTTTTTGGTCATCCAGAAGT | 50 °C | 800 | *Frohlich et al. (1999)* |
| | L2-N-3014: TCCAATGCACTAATCTGCCATATTA | | | |
| *P. aleyrodidarum 16S rDNA* | Pro-F: TGCAAGTCGAGCGGCATCAT | 59 °C | 1,000 | *Zchori-Fein & Brown (2002)* |
| | Pro-R: AAAGTTCCCGCCTTATGCGT | | | |
| *Cardinium 16S rDNA* | Ch-F: TACTGTAAGAAATAAGCACCGGC | 57 °C | 400 | *Zchori-Fein & Perlman (2004)* |
| | Ch-R: GTGGATCACTTAACGCTTTCG | | | |
| *Arsenophonus 23S rDNA* | Ars-F: CGTTTGATGAATTCATAGTCAAA | 60.5 °C | 900 | *Thao & Baumann (2004)* |
| | Ars-R: GGTCCTCCAGTTAGTGTTACCCAAC | | | |

and sequenced by IGE Biotechnology Co., Ltd (Guangzhou, China). The sequences were deposited in GenBank under accession numbers KP137471–KP137491 for *B. tabaci mtCO1* gene, KP201110–KP201126 for *P. aleyrodidarum* 16S *rDNA*, KP201103–KP201109 for *Arsenophonus* 23S *rDNA*, and KP201127–KP201134 for *Cardinium* 16S *rDNA* (Table 2).

## Sequence alignment and phylogenetic analysis

Sequence fragments were assembled using ContigExpress and aligned using the Clustal X 1.83 program (*Chenna et al., 2003*). The GenBank database was searched for homologous sequences of *mtCO1*, 16S *rDNA* and 23S *rDNA* using the basic local alignment search tool. Phylogenetic trees were constructed using MrBayes 3.2.1 (*Ronquist & Huelsenbeck, 2003*). The best-fit substitution model for each of the aligned sequences was selected with the program Modeltest 3.7 (*Posada & Crandall, 1998*). All the trees were constructed using the GTR+I+G model. The metropolis-coupled Markov chain Monte Carlo algorithm was conducted using four chains. Analyses were initiated with random starting trees, processed for $3 \times 10^6$ generations, and sampled every 1,000 generations. For the burn-in period, we discarded 100,000 generations. Posterior clade probabilities obtained from the analysis were used to assess nodal support. Tree information was visualized and edited using FigTree ver. 1.3.1 (http://tree.bio.ed.ac.uk/software/figtree/).

## RESULTS

### Molecular identification of *B. tabaci* individuals

Analyses of *mtCO1* sequences indicated that our 1,510 *B. tabaci* samples comprised of two invasive species (MEAM1 and MED) and three indigenous species (Asia II 1, Asia II 3, and China 1). Among the individuals tested, 36.4% (550/1,510) and 39.7% (600/1,510) were identified as MEAM1 and MED, respectively. The remained 4.6%, 9.3%, and 9.9% insects were identified as Asia II 1 (70/1,510), Asia II 3 (140/1,510), and China 1 (150/1,510), respectively (Fig. 1A; Table S1). Moreover, both the MEAM1 and MED whiteflies were widely distributed across China, whereas the three indigenous species

**Table 2 Haplotype information of whitefly, *P. aleyrodidarum*, *Arsenophonus*, and *Cardinium*.**

| Species | mt COI | | | | *P. aleyrodidarum* | | | | *Arsenophonus* | | | | *Cardinium* | | | |
|---|---|---|---|---|---|---|---|---|---|---|---|---|---|---|---|---|
| | n. | Re. seq. | Acc. no. | Per. (%) | n. | Re. seq. | Acc. no. | Per. (%) | n. | Re. seq. | Acc. no. | Per. (%) | n. | Re. seq. | Acc. no. | Per. (%) |
| MEAM1/B | 4 | BHZ-1 | KP137471 | 85.1 | 2 | BST-CA1 | KP201110 | 89.1 | | | | | 3 | BGD-CR1 | KP201127 | 92.5 |
| | | BST-2 | KP137472 | 8.2 | | BGW-CA2 | KP201111 | 10.9 | | | | | | BHC-CR2 | KP201128 | 2.5 |
| | | BSH-3 | KP137473 | 4.9 | | | | | | | | | | BSRF-CR3 | KP201129 | 5 |
| | | BJC-4 | KP137474 | 1.8 | | | | | | | | | | | | |
| MED/Q | 4 | QSQ-1 | KP137475 | 90.3 | 6 | QBJ-CA1 | KP201112 | 88.8 | 2 | QHS-AR1 | KP201103 | 89.7 | 2 | QSH-CR1 | KP201130 | 96.2 |
| | | QST-2 | KP137476 | 0.5 | | QAH-CA2 | KP201113 | 5.8 | | QHL-AR2 | KP201104 | 10.3 | | QSZ-CR2 | KP201131 | 0.4 |
| | | QHW-3 | KP137477 | 5.8 | | QSL-CA3 | KP201114 | 1.0 | | | | | | | | |
| | | QSZ-4 | KP137478 | 3.3 | | QSJ-CA4 | KP201115 | 2.0 | | | | | | | | |
| | | | | | | QSH-CA5 | KP201116 | 1.8 | | | | | | | | |
| | | | | | | QAH-CA6 | KP201117 | 0.5 | | | | | | | | |
| Asia II 3 | 5 | A3AJ-1 | KP137479 | 86.9 | 3 | A3ZW-CA1 | KP201118 | 86.2 | 1 | A3ZL-AR1 | KP201105 | 100.0 | 1 | A3ZL-CR1 | KP201132 | 100.0 |
| | | A3HW-2 | KP137480 | 1.5 | | A3ZL-CA2 | KP201119 | 9.2 | | | | | | | | |
| | | A3JN-3 | KP137481 | 0.8 | | A3GQ-CA3 | KP201120 | 4.6 | | | | | | | | |
| | | A3GQ-4 | KP137482 | 6.2 | | | | | | | | | | | | |
| | | A3ZL-5 | KP137483 | 4.6 | | | | | | | | | | | | |
| Asia II 1 | 3 | A1ZQ-1 | KP137484 | 81.4 | 2 | A1GQ-CA1 | KP201121 | 94.3 | 2 | A1ZL-AR1 | KP201106 | 92.3 | 2 | A1GQ-CR1 | KP201133 | 92.0 |
| | | A1GH-2 | KP137485 | 12.9 | | A1JS-CA2 | KP201122 | 5.7 | | A1GS-AR2 | KP201107 | 7.7 | | A1SL-CR2 | KP201134 | 8.0 |
| | | A1GH-3 | KP137486 | 5.7 | | | | | | | | | | | | |
| China 1 | 5 | CAJ-1 | KP137487 | 94.7 | 4 | CCY-CA1 | KP201123 | 2.7 | 2 | CCY-AR1 | KP201108 | 97.6 | | | | |
| | | CCY-2 | KP137488 | 2.0 | | CZS-CA2 | KP201124 | 90.7 | | CAJ-AR2 | KP201109 | 2.4 | | | | |
| | | CFW-3 | KP137489 | 0.7 | | CZH-CA3 | KP201125 | 5.3 | | | | | | | | |
| | | CJG-4 | KP137490 | 1.3 | | CZJ-CA4 | KP201126 | 1.3 | | | | | | | | |
| | | CGM-5 | KP137491 | 1.3 | | | | | | | | | | | | |
| Total | 21 | | | | 17 | | | | 7 | | | | 8 | | | |

**Note:**

*n.*, haplotype number; Re. Seq, representative sequence; Acc. no., accession number; Per, percentage in each cryptic species.
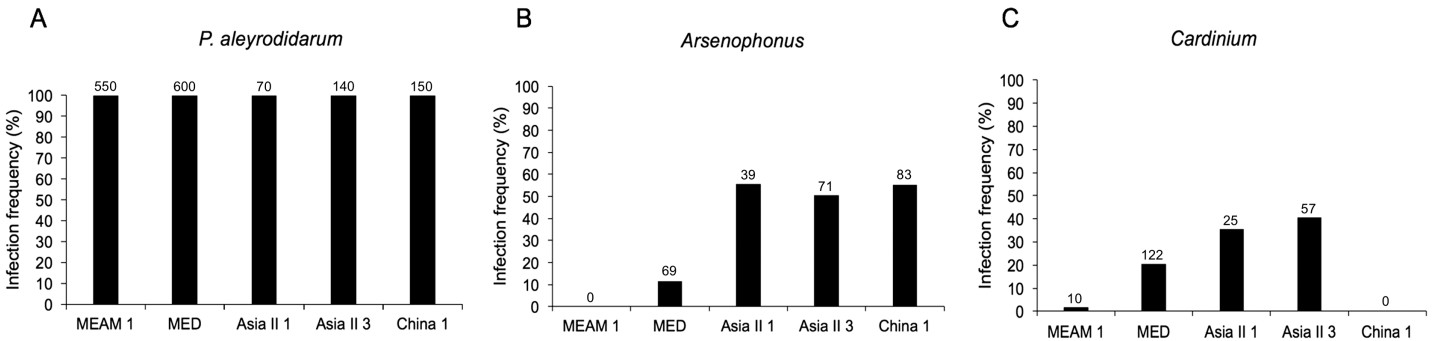

**Figure 2** **Infection frequency of endosymbionts in five *B. tabaci* cryptic species.** (A) *P. aleyrodidarum*; (B) *Arsenophonus*; (C) *Cardinium*. Number above bars indicate the number of infection.

(Asia II 1, Asia II 3, and China 1) were relatively less detected and mainly distributed in southeastern part of China (Fig. 1B).

## Prevalence of endosymbionts among five species of *B. tabaci*

As expected, *P. aleyrodidarum* was detected in all whitefly individuals and species (Fig. 2A); 21.7% (327/1,510) individuals harbored S-endosymbiont (*Arsenophonus* or *Cardinium*); 5.6% (84/1,510) whiteflies were co-infected with both, and the remained majority (72.8%; 1,099/1,510) lacked an infection with either of the two S-endosymbionts. Infection frequencies with S-endosymbionts varied across *B. tabaci* species. In detail, *Arsenophonus* was abundant in the indigenous species (Asia II 1, Asia II 3, and China 1), with infection rates ranging from 50.7% to 55.7%; however, it was infrequent in MED (11.5%) and MEAM1 (0.0%) (Fig. 2B). *Cardinium* was moderately common in MED, Asia II 1 and Asia II 3 populations with frequencies of 20.3–40.7%, but rarely detected in MEAM1 (0.5%) and not found in China 1 (Fig. 2C).

## Genetic diversity of *B. tabaci* and its endosymbionts

Aligned sequences from *B. tabaci* (813 bp, *mtCO1* gene), *P. aleyrodidarum* (886 bp, 16s *rDNA*), *Arsenophonus* (551 bp, 23S *rDNA*), and *Cardinium* (460 bp, 16S *rDNA*) were used to analyze the genetic variation of whitefly and its endosymbionts. The results showed that 21 haplotypes were identified in whiteflies based on *mtCO1* sequences, while 17, seven and eight symbiont haplotypes were defined based on analysis of *P. aleyrodidarum*, *Arsenophonus* and *Cardinium* sequences, respectively (Table 2). These haplotypes sequences were used to construct the phylogenetic trees.

## Phylogenetic analysis of *B. tabaci* and endosymbionts

Phylogenetic trees were constructed for *B. tabaci*, *P. aleyrodidarum*, *Arsenophonus,* and *Cardinium* based on *mtCO1* gene, 16s *rDNA*, 23S *rDNA,* and 16s *rDNA* sequences of haplotypes, respectively. We also pulled the related sequences from NCBI to explore the phylogenetic status of our haplotypes. The GenBank number of those related sequences could be found in phylogenetic trees. Based on the *mtCO1* gene, we found five distinct genetic groups of *B. tabaci*, which was corresponding with the five cryptic species

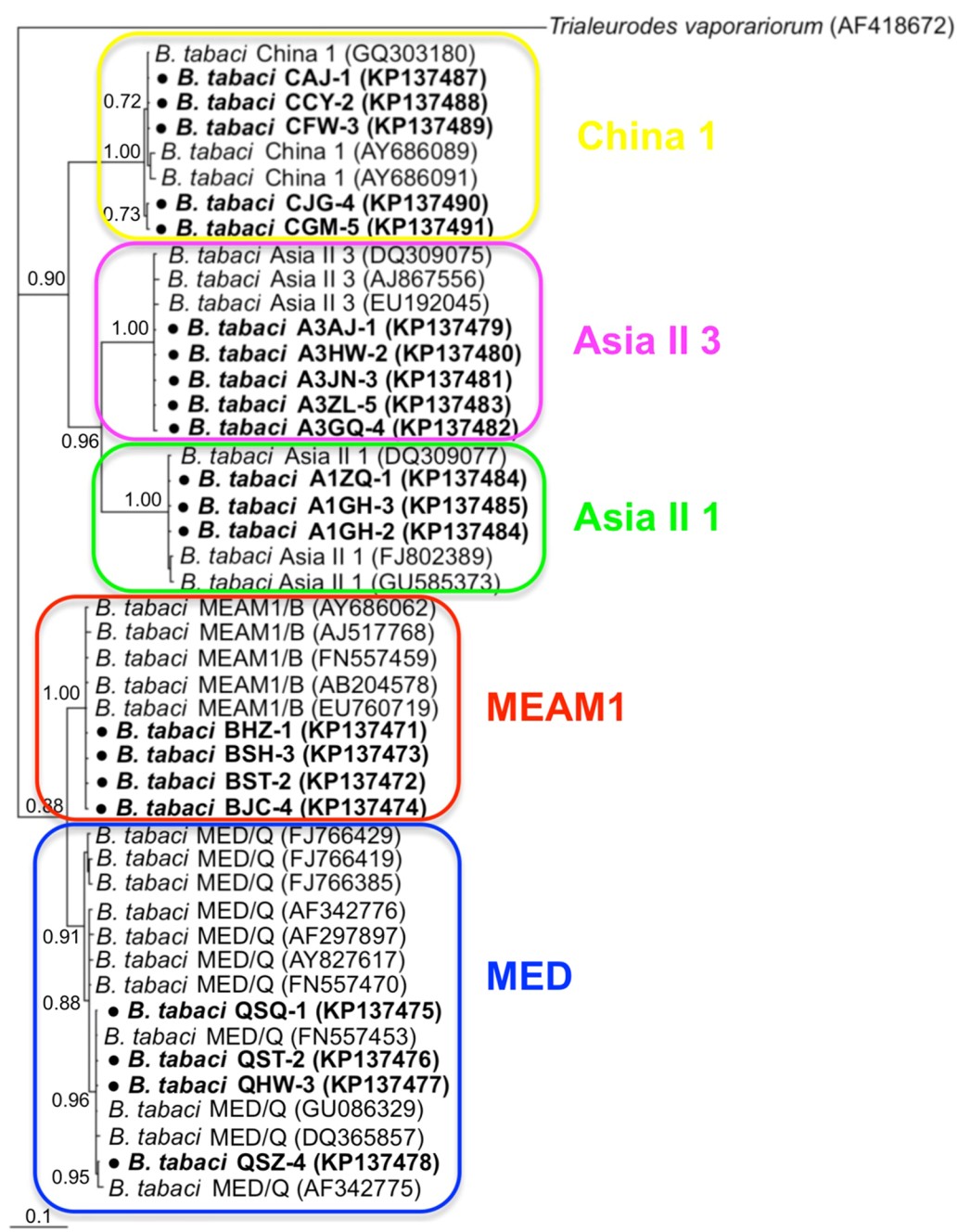

**Figure 3 The Bayesian phylogenetic tree of *B. tabaci* cryptic species based on *mtCOI* sequences.** The value beside the nodes are posterior probabilities. *Trialeurodes vaporariorum* (AF418672) is used as outgroup. Accession numbers for *mtCOI* sequences submitted to GenBank are KP137471–KP137491. All *mtCOI* sequences of *B. tabaci* cryptic species used in this study were clustered with other related references sequences from GenBank and their accession numbers are also indicated in the tree. Bold dots indicate the sequences from the present study.     

including the MEAM1, MED, Asia II 1, Asia II 3, and China 1 (Fig. 3). We can also find that all of our MED individuals belonged to Q1 subclade (*Chu et al., 2011*). For phylogenetic trees of endosymbionts, several distinct bacterial strains existed within

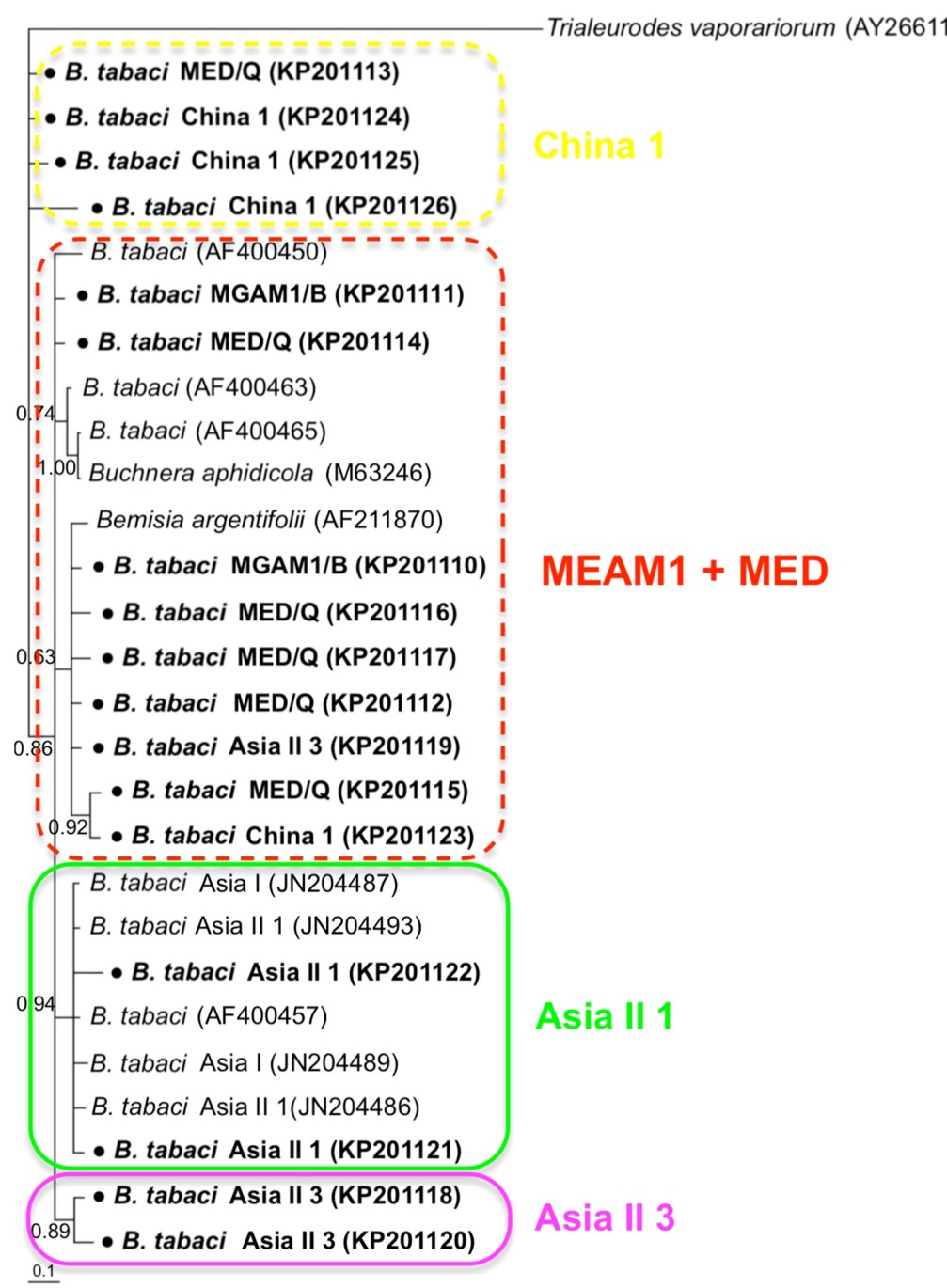

**Figure 4 The Bayesian phylogenetic tree of *P. aleyrodidarum* based on 16S *rDNA* sequences.** The value beside the nodes are posterior probabilities. *T. vaporariorum* (AY266113) is used as outgroup. Accession numbers for 16S *rDNA* sequences submitted to GenBank are KP201110–KP201125. All 16S *rDNA* sequences of *P. aleyrodidarum* used in this study were clustered with other related references sequences from GenBank and their accession numbers are also indicated in the tree. Bold dots indicate the sequences from the present study. Dotted boxes indicate imperfect cluster of each *B. tabaci* cryptic species.                              

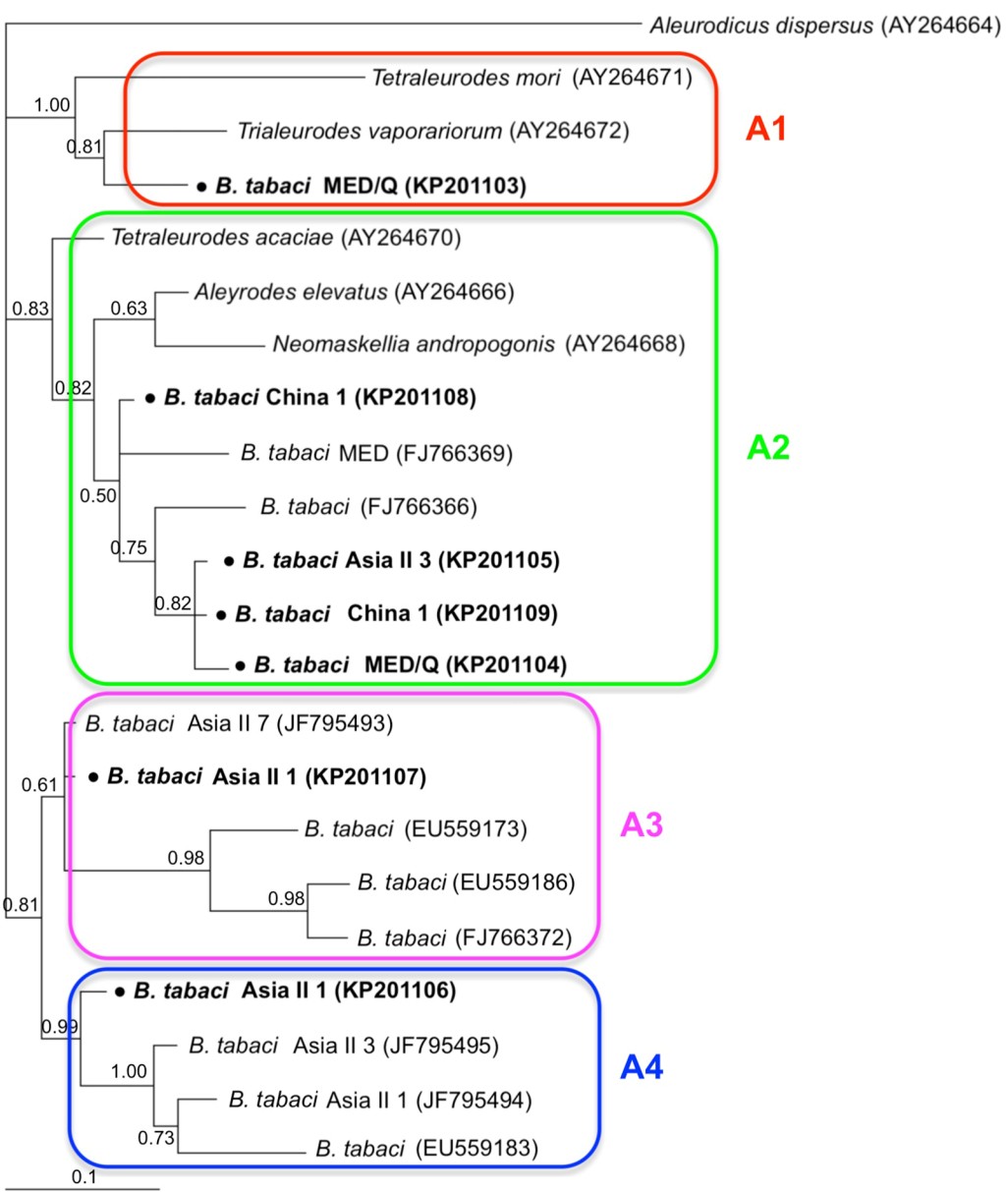

**Figure 5 The Bayesian phylogenetic tree of *Arsenophonus* based on 23S *rDNA* sequences.** The value beside the nodes are posterior probabilities. *Aleurodicus dispersus* (AY264664) is used as outgroup. Accession numbers for 23S *rDNA* sequences submitted to GenBank are KP201103–KP201109. All 23S *rDNA* sequences of *Arsenophonus* used in this study were clustered with other related references sequences from GenBank and their accession numbers are also indicated in the tree. A1–A4 indicate the four clusters.

individual bacterium. It is interesting that the phylogenetic tree of *P. aleyrodidarum* were similar to the *mtCO1* tree and exhibited four groups corresponding to MEAM1+MED, Asia II 1, Asia II 3, and China 1 clades to some extent (Fig. 4). While there are still several differences. For example, the sequences of *P. aleyrodidarum* of China 1 were not well clustered and *P. aleyrodidarum* from *B. tabaci* species MED (KP201113) was also present in that China 1 clade. In addition, the MEAM1+MED clade (Fig. 4) contained

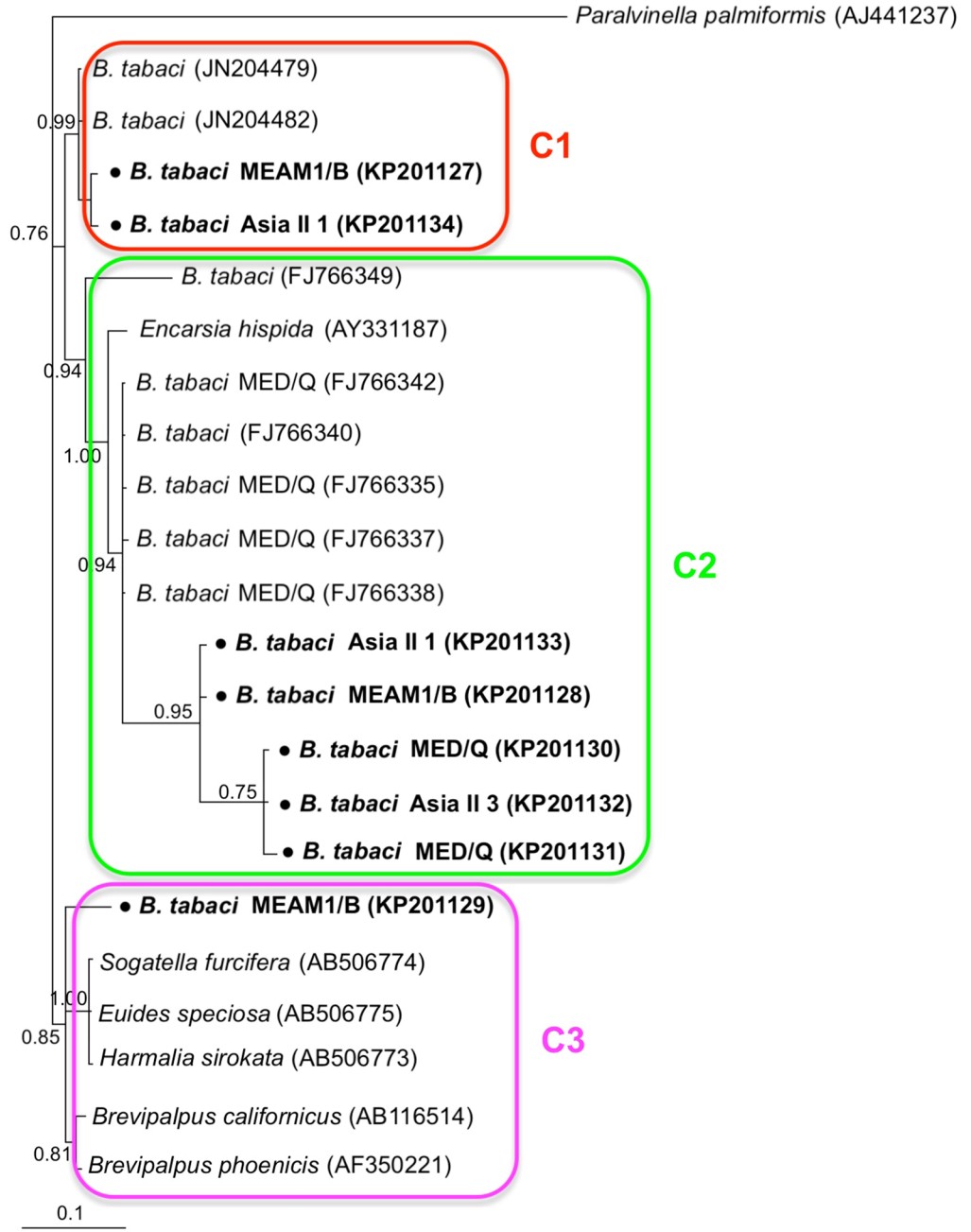

**Figure 6 The Bayesian phylogenetic tree of *Cardinium* based on 16S *rDNA* sequences.** The value beside the nodes are posterior probabilities. *Paralvinella palmiformis* (AJ441237) is used as outgroup. Accession numbers for 16S *rDNA* sequences submitted to GenBank are KP201127–KP201134. All 16S *rDNA* sequences of *Cardinium* used in this study were clustered with other related references sequences from GenBank and their accession numbers are also indicated in the tree. C1–C3 indicate the three clusters.

sequences of *P. aleyrodidarum* from both MEAM1 and MED; however, it was separated into two distinct clades in the *B. tabaci* tree (Fig. 3). In contrast, the phylogenetic trees for *Arsenophonus* and *Cardinium* were totally incongruent with the *mtCO1* tree, exhibiting four (A1–A4) and three groups (C1–C3), respectively (Figs. 5 and 6).

## DISCUSSION

In this study, we conducted an extensive screening of *B. tabaci* for the presence of one P-endosymbiont and two common S-endosymbionts, along with phylogenetic analyses of these symbionts to compare with host species from the cryptic *B. tabaci* complex. The reason we chose *Arsenophonus* and Cardinium as representatives of S-endosymbionts because they are the very common S-endosymbionts in whiteflies, and *Wolbachia* has been thoroughly investigated in the study of *Bing et al. (2014)*. The results showed that P-endosymbiont *P. aleyrodidarum* was detected in all whitefly individuals while S-endosymbionts infection were varied among species. The variation in the prevalence of endosymbionts could be influenced by numerous factors such as host, environmental conditions, geographical location or even climate (*Chu et al., 2011*; *Karut & Tok, 2014*; *Morag et al., 2012*; *Skaljac et al., 2010*). In our study, *Arsenophonus* was abundant in Asia II 1, Asia II 3, and China 1 species but absent in the invasive species MEAM1, which is exactly consistent with previous studies (*Bing et al., 2013*; *Karut et al., 2017*); *Cardinium* was present in the MED, Asia II 1 and Asia II 3 species (20.3–40.7%) but was rarely detected in MEAM1 and not detected in China 1. Taken together, it seemed that these two S-endosymbionts had high prevalence in native species rather than invasive species, which is consistent with another S-endosymbiont *Wolbachia* but in contrast to *Hamiltonella*; *Hamiltonella* was found abundant in invasive species rather than native species (*Bing et al., 2013*).

Previous studies showed that *B. tabaci* could be co-infected with particular pairs of S-endosymbionts, including *Rickettsia* and *Hamiltonella*, *Hamiltonella* and *Cardinium*, or *Rickettsia* and *Arsenophonous*; others were less common, such as *Cardinium* and *Rickettsia*, *Hamiltonella* and *Arsenophonous*, *Cardinium* and *Wolbachia*, and even three or four endosymbionts together (*Gueguen et al., 2010*; *Karut & Tok, 2014*; *Marubayashi et al., 2014*; *Pan et al., 2012*; *Skaljac et al., 2010*). In our study, we found evidence for a low rate (5.6%) of co-infection with *Arsenophonous* and *Cardinium* in four *B. tabaci* species (MEAM1 was the exception since *Arsenophonous* was not detected in this species), which has also been reported before (*Bing et al., 2013*; *Chu et al., 2011*; *Parrella et al., 2014*; *Zchori-Fein, Lahav & Freilich, 2014*). However, the reason of so few rate of co-infection by *Arsenophonous* and *Cardinium* is that *Arsenophonous* and *Cardinium* are potential reproductive manipulators that compete for resources inside the bacteriocytes, thus compromising the fitness of host (*Gottlieb et al., 2008*). We have one plausible explanation for the co-infection status, that is *Cardinium* is not restricted to the bacteriocytes (*Skaljac et al., 2010*), and perhaps the non-overlapping niche makes co-infection of *Arsenophonous* and *Cardinium* possible. In addition, the co-infection symbiont system in whiteflies may indicate the roles of dual endosymbionts: work as important mutually dependents to provide full complement of nutrients to their host (*Rao et al., 2015*) or affect the fitness and biology of the *B. tabaci* (*Ghosh et al., 2018*).

Our phylogenetic analyses indicated that *B. tabacia* mt*CO1* sequences could be assigned to five distinct clades, which conformed to existing MEAM1, MED, Asia II 1, Asia II 3,

and China 1 clades. Similarly, the sequences of the P-endosymbionts *P. aleyrodidarum* were assigned to their own clade and the phylogeny was similar with that of *B. tabaci* genetic groups to some extent. This may potentially indicate an ancient infection followed by vertical transmission and subsequent co-evolutionary diversification (*Baumann, 2005*). Meanwhile, it is important to note that the correlation was not perfect: sequences of *P. aleyrodidarum* from MEAM1 and MED were assigned to the same clade instead of the two distinct clades presented in the mt*CO1* tree. The reason could be the dissemination of the MEAM1 and MED species; furthermore, the spread of these two invasive species in China has been frequently associated with founder effects that fix specific mtDNA variation(s) along with particular endosymbionts (*Chu et al., 2011*; *Gueguen et al., 2010*). Taken together, although there was similarity between the two trees of *P. aleyrodidarum* and *B. tabaci*, the genetic variation of primary symbiont might not be an ideal reflecting the genetic variation of the cryptic *B. tabaci*.

The S-endosymbionts, *Arsenophonus* and *Cardinium,* both showed a lack of congruence with the *B. tabaci* mt*CO1* phylogram. This is consistent with the finding from *Ahmed et al. (2013)*, who provided evidence for horizontal transmission of S-endosymbionts in the *B. tabaci* cryptic species complex based on phylogenies studies. There are substantial phylogenetic evidences showing that S-endosymbionts such as *Wolbachia* and *Arsenophonus*, undergoing horizontal transfer among host arthropod species (*Ahmed et al., 2013*; *Chrostek et al., 2017*; *Kolasa et al., 2017*; *Li et al., 2017*; *Russell et al., 2003*; *Vavre et al., 1999*). In some cases, the mechanisms for horizontal transmission of S-endosymbionts are already known, including transferring through parasitoid wasps (*Ahmed et al., 2015*; *Gehrer & Vorburger, 2012*), plants (*Caspi-Fluger et al., 2012*; *Li et al., 2017*) or even sexual transmission (*Moran & Dunbar, 2006*). Therefore, the potential horizontal transfer of S-endosymbionts in our samples could be one or combination of the above ways.

In summary, this study reported the varied prevalence of three endosymbionts within five *B. tabaci* cryptic species. The P-endosymbiont *P. aleyrodidarum* was detected in all whitefly individuals; the S-endosymbionts *Arsenophonus* was abundant in native species while *Cardinium* was common in the invasive species. In addition, the phylogenetic relationships between endosymbionts and their hosts *B. tabaci* probably indicated the vertical transmission and co-evolution of *P. aleyrodidarum* and *B. tabaci*; meanwhile, horizontal transfer of *Arsenophonus* and *Cardinium* may happen in our collecting samples. Our study not only reported current infection status of endosymbionts within *B. tabaci* populations in China, but also demonstrated that S-endosymbionts genetic variation may not reflect host genetic variation and should not be used to infer taxonomic relationships within the host species complex. If funding allows, more endosymbionts should be investigated and future investigations could be the contribution of endosymbionts to invasiveness, population expansion, or even competitiveness of whitefly species.

## ACKNOWLEDGEMENTS

We express our deep gratitude to the Testing Center of Yangzhou University. We sincerely thank Dr. Kerry M. Oliver from Department of Entomology, University of Georgia for polishing the manuscript and providing valuable comments.

### Funding

This study was supported by the Special Fund for Agro-scientific Research in the Public Interest of China (Nos. 201303019, 200803005). The funders had no role in study design, data collection and analysis, decision to publish, or preparation of the manuscript.

### Grant Disclosures

The following grant information was disclosed by the authors:
Special Fund for Agro-scientific Research in the Public Interest of China: 201303019, 200803005.

### Competing Interests

The authors declare that they have no competing interests.

### Author Contributions

- Xiao-Tian Tang analyzed the data, contributed reagents/materials/analysis tools, prepared figures and/or tables, authored or reviewed drafts of the paper, approved the final draft.
- Li Cai analyzed the data, contributed reagents/materials/analysis tools, prepared figures and/or tables, authored or reviewed drafts of the paper, approved the final draft.
- Yuan Shen performed the experiments, authored or reviewed drafts of the paper, approved the final draft.
- Yu-Zhou Du conceived and designed the experiments, authored or reviewed drafts of the paper, approved the final draft.

### DNA Deposition

The following information was supplied regarding the deposition of DNA sequences:
GenBank accession numbers: KP137471–KP137491 for the *B. tabaci* mtCO1 gene, KP201110–KP201126 for the *P. aleyrodidarum* 16S rDNA, KP201103–KP201109 for Arsenophonus 23S rDNA, and KP201127–KP201134 for Cardinium 16S rDNA.

### Data Availability

The raw data are provided in a Supplemental File.

### Supplemental Information

Supplemental information for this article can be found online at http://dx.doi.org/10.7717/peerj.5516#supplemental-information.

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
