# Peer review of "Diversity and evolution of the endosymbionts of Bemisia tabaci in China"

_PeerJ, doi:10.7717/peerj.5516_

## Round 0.1 · original submission · Major Revisions

Four experts have now reviewed your paper and there are several areas of improvement. Please respond to each concern, especially the request to include the data for the other key endosymbionts.

Reviewer 1 ·

Basic reporting

The research is straight forward and not complicated. The manuscript is nicely written. How ever, the authors should pay more attention to some details.
For instance:
1 The genus and species names in literature references should be italic.
2 In figures 4, 5, 6, the names of symbionts should be explained well. The names of insect hosts should be italic. The authors should show full genus names of the insect host for the first time.
3 The new identified symbionts should have host information like "Bemisia tabaci China1 XXX".
4 The bold symbols had better have clearer notes on phylogenetic figures.
5 Line 69, Orientia is not the newly found S-symbiont in B. tabaci. It should be checked again.

Experimental design

I am curious why the authors only investigated 2 of the 7 secondary symbionts in B. tabaci. Do they have any special interest? They may be discussed in the revised manuscript.

Validity of the findings

1 It's unprofessional to show "MEAM1/B" in the abstract. Full names should be shown for the first time shown in a paper. The biotype B or Q is not helpful here and can be deleted.
2 Figure 3, Bemisia samples should be noted so that they are different from T vaporariorum.
3 In the abstract, Line 27- 30, "phylogenetic analyses revealed that the P. aleyrodidarum and mtCO1 (mitochondrial cytochrome oxidase 1) phylograms were highly identical and corresponding with the five distinct cryptic species clades, indicating an ancient infection followed by strict vertical transmission and subsequent co-evolutionary diversification" . But, this is not supported well by data shown in Figure 3 and Figure 4. The phylogeny of Portiera is not clustered well.
It should be better to build a figure that shows whether the two trees have evolutionary consistence.

Additional comments

The authors should discuss more on the findings. Although the text was corrected by native speakers, the figures and details were obviously need improvement.

Reviewer 2 ·

Basic reporting

No comment

Experimental design

No comments

Validity of the findings

No comments

Additional comments

The research paper “Diversity and evolution of the endosymbionts of Bemisia tabaci in China” by authors Tang and Cia et al., describes the prevalence and diversity of one primary and two facultative symbionts in china. Authors samples 1501 B. tabaci individuals which belong to two invasive species (MEAM1 and MED) and three indigenous species (Asia II1, Asia II 3 and China 1). Though the authors collected good number of samples at 71 geographical locations across the China, my main concern of this study is why authors focused only on primary and two secondary endosymbionts (Arsenophonus and Cardinium), What about other secondary endosymbionts?. At present at least 40 morphologically indistinguishable species were reported from B. tabaci cryptic species (De Barro et al., 2011, Dinsdale et al., 2010 and Hu et al., 2017); authors should update their species information by citing above reference papers. As other studies have already shown endosymbiont diversity at an intra-species level, this study does not bring much information as other studies. Authors might have focused on all endosymbionts to study the strict vertical/horizontal transmission and co-evolution of endosymbionts in their samples collected. As a summary, no significant novel information are presented in this manuscript, except collecting and screening huge number of samples across the China. Also authors should have been performed phylogenetic analysis more than one more marker to represent evolutionary relationships of endosymbionts and B. tabaci. I would have made an interim decision for “major revision” if authors focused at least known five endosymbionts and more than one marker to study the evolutionary relationships.

Reviewer 3 ·

Basic reporting

no comment

Experimental design

no comment

Validity of the findings

no comment

Additional comments

The manuscript by Tang and co-workers represents a systematic survey of three symbionts (Portiera, Arsenophonus and Cardinium) of Bemisia tabaci in China. The authors provide a valuable survey with wide coverage. The study is valuable for providing wide coverage and it is an effective supplement to the present situation of symbionts of B. tabaci in China.

Following, my comments on the manuscript:

In this study, the author rarely discussed other s-symbionts, such as Wolbachia and Hamiltonella, the later distributes widely in invasive species MED/Q and MEAM1/B.

As we known, some biotypes of whitefly have co-infection symbiont system. The dual endosymbionts showed important mutually dependent to provide a full complement of nutrients to their host whiteflies (ref.1), and effect on the fitness and biology of the Bemisia tabaci (ref.2). In the discussion, the relationship between the different symbionts might be discussed.
Ref.1 Rao Q, PA Rollat-Farnie, DT Zhu, et al. Genome reduction and metabolic complementation of the dual endosymbionts in the whitefly Bemisia tabaci,BMC Genomics,2015, 16:226.
Ref.2 Ghosh S, Bouvaine S, Richardson S C W, et al. Fitness costs associated with infections of secondary endosymbionts in the cassava whitefly species Bemisia tabaci. Journal of Pest Science, 2018, 91:17-28.

From figure 3 and figure 4, the relationship of clusters between P. aleyrodidarum and cryptic species is not obviously clear. The genetic variation of primary symbiont might not be an ideally reflect the genetic variation of this cryptic B. tabaci.

Line 155-157: “The results showed that 21 haplotypes were identified in whiteflies based on mtCO1 sequences, while 17, 7 and 8 symbiont haplotypes were defined based on analysis of P. aleyrodidarum, Arsenophonus and Cardinium sequences, respectively”. How to identify the haplotypes of whiteflies and symbiont haplotypes? What these claim are based on?

Line 170-172: “It is interesting that the phylogenetic tree of P. aleyrodidarum were highly similar to the mtCO1 tree and exhibited four groups corresponding to MEAM1/B+MED/Q, Asia II 1, Asia II 3 and China 1 clades (Fig. 4).” From figure 4, the sequences of P. aleyrodidarum of China1 are not clustered.

Line 222-224: “furthermore, the spread of these two invasive species in China has been frequently associated with founder effects that fix specific mtDNA variation(s) along with particular endosymbionts”, the sentence is not very clear.

Figure 2B and 2C: the total number of each cryptic B. tabaci on the bar graph is not necessary, because all samples are mentioned. The number of infection should more interest for audience.

Reviewer 4 ·

Basic reporting

The manuscript by Tang et al. investigated the distribution frequency of the primary and two secondary endosymbionts of five species of Bemisia tabaci from China, and then outlined the co-evolution of the mutualists by phylogenetic analysis.

Experimental design

no comment

Validity of the findings

The data seem to provide some new insights into the endosymbiotic status and evolutionary relationships in whitefly.

Additional comments

1. In B. tabaci whitefly, seven secondary endosymbionts have been identified and many of them have abundant distribution, why the authors only concerned Arsenophonus and Cardinium rather than others in this study?

2. Phylogenetic trees of Portiera and Cardinium were constructed with 16S rDNA sequences, while 23S rDNA was used for Arsenophonus. I wonder why different genes were chosen for phylogenetic analysis in different bacteria.

3. Line 34-35: The vertical and horizontal transmission of symbionts was speculative and no direct evidences were provided in this study, please check your expression and make it unambiguous.

4. Line 195-200: The interpretation of low frequency of symbionts was appeared to be not sound. Reproduction manipulation usually facilitates spread of symbionts in the hosts and sometimes may benefit their hosts, such as high proportion of female. Besides, the assumed negative effects of the symbionts can’t explain why those bacteria have high prevalence in native species but low frequency or absence in invasive species.

5. Line 206-208: The diversity of symbionts has been well studied in B. tabaci, and the co-infection of the Cardnium and Arsenophonous has also been reported in previous studies (Bing et al., 2013; Chu et al., 2011; Parrella et al., 2014; Zchori-Fein et al., 2014).

6. The text of scientific names of organisms should be italicized and the style of references should be consistent.

7. In figure 1B, the native species can be labeled separately with different symbols.

8. In table 2, the total number of mtCOI haplotypes should be 21, please check and revise. How did authors calculate the percentage of each haplotype, by sequencing all samples or other methods?

---

## Round 0.2 · accepted · Accept

Thank you very much for revising your manuscript. Both myself and the reviewers agree it is acceptable for publication.

All the best,
Laura Boykin

# Reviewer 3 ·

Basic reporting

no comment

Experimental design

no comment

Validity of the findings

no comment

Additional comments

no comment

Reviewer 4 ·

Basic reporting

no comment

Experimental design

no comment

Validity of the findings

no comment